# CRISPR-Based Approaches for the High-Throughput Characterization of Long Non-Coding RNAs

**DOI:** 10.3390/ncrna7040079

**Published:** 2021-12-13

**Authors:** Joshua Hazan, Assaf Chanan Bester

**Affiliations:** Faculty of Biology, Technion, Technion City, Haifa 3200003, Israel; joshuah@campus.technion.ac.il

**Keywords:** ncRNA, lncRNA, CRISPR, CRISPRi, CRISPRa, enhancer, Cas9, Cas13

## Abstract

Over the last decade, tens of thousands of new long non-coding RNAs (lncRNAs) have been identified in the human genome. Nevertheless, except for a handful of genes, the genetic characteristics and functions of most of these lncRNAs remain elusive; this is partially due to their relatively low expression, high tissue specificity, and low conservation across species. A major limitation for determining the function of lncRNAs was the lack of methodologies suitable for studying these genes. The recent development of CRISPR/Cas9 technology has opened unprecedented opportunities to uncover the genetic and functional characteristics of the non-coding genome via targeted and high-throughput approaches. Specific CRISPR/Cas9-based approaches were developed to target lncRNA loci. Some of these approaches involve modifying the sequence, but others were developed to study lncRNAs by inducing transcriptional and epigenetic changes. The discovery of other programable Cas proteins broaden our possibilities to target RNA molecules with greater precision and accuracy. These approaches allow for the knock-down and characterization of lncRNAs. Here, we review how various CRISPR-based strategies have been used to characterize lncRNAs with important functions in different biological contexts and how these approaches can be further utilized to improve our understanding of the non-coding genome.

## 1. Introduction

A large proportion of the human genome encodes various types of regulatory elements and non-coding genes. Of these, the long non-coding RNA (lncRNA), defined as a transcript that is longer than 200 nt with low or no protein-coding potential [1,2], is one of the largest—and arguably among the most poorly characterized—classes of non-coding RNA genes [3,4,5]. A major boundary to characterizing and identifying lncRNAs is their relatively low expression, high cell-type specificity, and poor sequence conservation across species [4,5,6,7,8]. While the majority of lncRNAs that have been identified until now remain uncharacterized, some transcripts have been associated with a wide range of cellular functions and biological processes [2,5]. Some lncRNAs, such as X-inactive specific transcript (XIST) [9] and non-coding RNA activated by DNA damage (NORAD) [10,11,12], are vital for normal cellular function. Others have been implicated in diseases such as cancer, including metastasis associated lung adenocarcinoma transcript 1 (MALAT1) [13] and plasmacytoma variant translocation 1 (PVT1) [14].

Very few of the abundant lncRNAs in the human genome have been properly functionally classified and it remains difficult to accurately predict functional lncRNAs computationally; therefore, high-throughput assays are important for characterizing functional lncRNAs in different tissues and contexts [15,16]. The bacterial clustered regularly interspaced short palindromic repeats (CRISPR)-CRISPR associated protein 9 (Cas9) nuclease system was recently discovered as a highly adaptable strategy for genetic manipulation [17] and has been used for genetic manipulation in mammalian cells for a wide variety of experiments [18,19]. Briefly, the CRISPR-Cas9 system works through the guiding of the Cas9 protein to a target sequence through a guide RNA (gRNA) that is comprised of a CRISPR-RNA (crRNA) that is base paired to a trans-activating crRNA (tracrRNA) that is required to process the mature gRNA. The Cas9 directly identifies a short, three-nucleotide sequence in the DNA that is known as the protospacer-adjacent motif (PAM), leading to the Cas9 opening the DNA; the gRNA then generates a stable R-loop with the target site [20]. Finally, the nuclease activity of Cas9 induces a DNA double strand break (DSB). For genetic manipulation experiments, this is generally combined into a single guide RNA (sgRNA) that includes both RNA components. Upon cleavage, the DNA repair machinery is recruited to the DSB, often inducing point mutations or frameshift mutations at the target site to functionally knock out the target protein (CRISPR-ko) [17,18,19]. CRISPR has been further adapted for the manipulation of gene expression without modifying the genome through the use of a nuclease-dead Cas9 (dCas9), which binds the target site but does not cleave the DNA [21]. This system has been adapted for both gene inhibition (CRISPRi) [22] and activation (CRISPRa) [23,24]. These methods, in addition to classical CRISPR-ko, have been adapted in a wide range of strategies and have been applied for various high-throughput screens of functional lncRNAs in many different cell types to improve the functional characterization of lncRNAs (Figure 1) [25,26,27,28]. Novel Cas variants, notably Cas12 [29] and Cas14 [30], may lead to further improvements in gene editing technologies and allow for multiplex approaches due to the differences in their mechanisms of action. In addition, the newly discovered Cas13 enzyme, which binds and modifies the RNA rather than the DNA, shows potential for high-throughput lncRNA analysis at the transcriptional level [31,32]. Finally, non-CRISPR-based approaches, such as RNA interference (RNAi) [33] and antisense oligonucleotides (ASOs) [34], have been used for high throughput lncRNA screens, and can be complementary approaches to CRISPR [35,36]. Table 1 summarizes the major approaches that are used for high-throughput analysis of regulatory elements and lncRNAs.

In this review, we summarize different CRISPR-based approaches, focusing on high-throughput applications to study non-coding genes, DNA regulatory elements, and lncRNAs. We further discuss the advantages and limitations of the different approaches, as well as emerging technologies and future directions for clarifying the biological roles of lncRNAs in basic and translational studies.

### 1.1. Proposed and Proven Functions of lncRNAs

There are many lncRNAs with characterized functions, both in normal cell activity and in disease states. Here, we briefly explain four of the most comprehensively characterized and well known lncRNAs with a major function.

The first major lncRNA to be discovered was XIST, which controls the X-inactivation system in mammals. X-inactivation is an important mechanism for dosage compensation in mammals to balance the expression of X-linked genes between females and males. X-inactivation begins in the early female embryo, and the inactive X (Xi) is seemingly chosen at random. XIST was first discovered as a 15–17-kb lncRNA transcribed from the X inactivation center (Xic) locus in Xi only [63]; it was subsequently found to “paint” the entire chromosome and recruit factors to alter chromatin structure and modify epigenetic markers such as methylation to prevent transcription [64,65,66]. Since its discovery, much research has been done to determine the functional elements of XIST and how the transcript establishes and propagates Xi; this has been reviewed in detail elsewhere [9,67]. Due to its vital role in dosage compensation, XIST is the most well characterized lncRNA in humans.

XIST is a classic example of a cis-acting lncRNA—lncRNAs that work in close proximity to their loci [68]. However, other lncRNAs act at a distance from their transcription site—trans-acting lncRNAs. NORAD is an example of such a lncRNA.

NORAD is a recently discovered, highly conserved lncRNA that is important for maintaining genomic stability [10,11]. Its inactivation triggers aneuploidy in otherwise stable human cells, and it was shown to sequester the Pumilio proteins PUM1 and PUM2; in the absence of NORAD, these proteins repressed factors that are related to mitosis, DNA repair, and DNA replication, leading to major genomic instability [10]. NORAD is comprised of repetitive units including several Pumilio binding sites; it binds PUM1 and PUM2 to regulate the mRNA levels of Pumilio target genes, many of which are associated with chromosome segregation during cell division [11]. Other interaction partners for NORAD have been discovered, notably SAM68, an RNA-binding protein that is important for the recruitment of PUM2 to NORAD [12]. In addition to those with roles in normal cell function, many lncRNAs have been identified with a role in disease states, especially cancer.

Among the first cancer-associated lncRNAs to be identified was MALAT1, also known as nuclear enriched abundant transcript 2 (NEAT2). It was initially found to be strongly associated with metastasis in non-small cell lung cancer (NSCLC) [69]. Unusually for an lncRNA, it is highly conserved across species [70]. The transcript is enriched in the nucleus, specifically within nuclear speckles [70,71], and functions in modulating pre-mRNA splicing in nuclear speckles [71]. However, the role of MALAT1 in cancer is controversial, as conflicting studies have shown that it can act as an oncogenic factor [72,73,74] or as a tumor suppressor [75,76]. It is possible that MALAT1 has different effects in different cancer types, but the true role of the transcript in cancer remains undecided. The roles of MALAT1 in normal cell function and in cancer have been covered in detail elsewhere [13].

The lncRNA PVT1 was found to be related to the MYC oncogene, and both are located within the 8q24.21 genomic region; PVT1 undergoes a copy number increase in many human cancers [14]. It was originally found to be associated with ovarian and breast cancer progression and reduced survival, and its silencing by short interfering RNA (siRNA) led to reduced growth and increased apoptosis in cell lines with *MYC* and PVT1 gain/overexpression [77]. Tseng et al. [78] then found that a gain of PVT1 was required for MYC up-regulation in vitro and that PVT1 copy number was increased in nearly all MYC-driven cancers. Li et al. [79] then investigated the potential interactions of PVT1 with the microRNA miR-152, which is consistently down-regulated in cancers with increased PVT1. They showed that PVT1 has three potential binding sites for miR-152 and that PVT1 may act as a “sponge” to inhibit miR-152 in gastric cancer. Similarly, PVT1 was found to accelerate the progression of NSCLC [80]. However, the promoter of PVT1 appears to have the opposite effect, as additional studies have shown that the PVT1 promoter is a tumor suppressor element; silencing the PVT1 promoter was shown to enhance MYC expression and cell growth in several breast cancer cell lines rather than inhibiting it as prior evidence would suggest [81].

### 1.2. High-Throughput Reverse Genetic Screening

As thousands of lncRNAs have been identified and only a fraction of them have been characterized, there is a need for robust methods for the identification of functional lncRNAs in different biological contexts. Reverse genetics is a strategy to identify the function of a gene or genetic locus in which a specific sequence is targeted by approaches that affect its sequence or expression levels. The outcome of this manipulation is measured as the phenotype. Classically, the phenotype can be cell-based (e.g., changes in cell fitness) or reporter-based (e.g., the expression of a fluorescent protein). More recently, using single cell RNA sequencing (scRNA-seq), the phenotype of genetic screens can be measured in terms of changes in transcription levels. Reverse genetic screening can be performed as an array or in pooled approaches. In arrayed screening, cells are plated in different wells and specific genetic manipulations occur in each well. In pooled screening, genetic manipulation occurs randomly on a pool of cells. In most cases, the experiment is designed so that each cell will be manipulated only once. After exposing the cells to some form of selective pressure (in cell-based phenotypes) or measurement and sorting of the reporter (in reporter-based phenotypes), next generation sequencing (NGS) is used to recover the specific genetic manipulation in the group of interest and the phenotype is compared to that in control cells. While arrayed screening is usually more limited in terms of the number of genetic manipulations, pooled screening can be performed with up to hundreds of thousands of different manipulations in one experiment and is, therefore, highly relevant for functional lncRNA screens. There are different approaches to reverse genetic screening; however, the most well-known are based on short hairpin RNA (shRNA) and CRISPR. Although beyond the scope of this manuscript, detailed reviews about the experimental design of reverse genetic screens can be found elsewhere [82,83,84]. An overview of the process is shown in Figure 2.

## 2. CRISPR-Based Approaches to Study lncRNAs

### 2.1. Using CRISPR-Ko to Directly or Indirectly Knock Out lncRNAs

The programmable nature of CRISPR-Cas9 was first confirmed in bacteria by the team of Charpentier and Doudna [17]. In 2013, CRISPR was first adapted as a tool for gene editing in eukaryotic cells by the teams of both Feng Zhang [18] and George Church [19]. CRISPR-Cas9 was initially developed as a tool for the knock-out (CRISPR-ko) and knock-in of protein-coding genes. The Cas9 protein has endonuclease activity that generates DSBs to activate the DNA damage response, potentially leading to genetic changes [17]. In mammalian cells, the non-homologous end joining (NHEJ) mechanism is the main DNA repair pathway. NHEJ involves DNA processing and ligation that usually leads to a 1–3-nucleotide insertion or deletion (indel) [16,26], though larger deletions are not uncommon [18,19]. Often, these indels lead to frameshift mutations that may result in a premature stop codon and the activation of nonsense-mediated decay (NMD), leading to the degradation of the mRNA. Clearly, this mechanism does not apply to non-coding genes and regulatory elements that are lacking an open reading frame (ORF). Nevertheless, some non-coding elements are relatively small and their function relies heavily on their sequence (e.g., enhancers, miRNA, snoRNA). Thus, these sites are often sensitive to the short indels that are generated by a single Cas9-gRNA complex.

Enhancers are genomic regulatory elements that bind specific transcription factors (TFs), which, in many cases, overlap non-coding genes such as enhancer RNA and lncRNAs [85]. Therefore, studying enhancer regions often involves the study of ncRNAs. Korkmaz et al. [37] performed a large-scale CRISPR-Cas9 screen with a library that was designed to target enhancers. While specific enhancers can bind multiple TFs, the binding site for a specific TF is only a few base pairs in length. This limits the number of sgRNAs that can effectively target these sites, and many sites cannot be targeted by a simple sgRNA; instead, more advanced strategies are required. Nevertheless, the authors were able to identify functional enhancers using a library of sgRNAs that were designed to target previously identified p53 binding sites.

While Korkmaz et al. [37] used prior knowledge to design their sgRNA library, another strategy for enhancer identification, in which there is no—or limited—prior knowledge regarding a specific TF, is to tile a suspected genomic region with as many sgRNAs as possible. Canver et al. [38] demonstrated this approach by tiling suspected enhancers of BCL11A. Using 650 sgRNAs tiling a total of 4000 bp, the authors were able to map sequences that regulate BCL11A expression, both positively and negatively, at a high resolution. Since single nucleotide polymorphisms (SNPs) and expression quantitative trait loci (eQTLs) are common in enhancers and other non-coding regions, a CRISPR-based approach may be useful to assess the function of these sites.

Introducing short indels using a single sgRNA has limitations when targeting short genomic regions (e.g., enhancers), as well as for genes whose function is not heavily dependent on their sequence (e.g., lncRNAs). Therefore, an alternative strategy is to generate a genomic deletion. Gasperini et al. [43] utilized an approach in which paired sgRNAs (pgRNAs) were designed to perform thousands of kilobase-scale tiled deletions in a region surrounding the housekeeping gene HPRT1 through a technique called ScanDel. However, their deep genomic deletion screen failed to identify any single enhancer that was critical for HPRT1 expression. More recently, the same group developed a different screening approach that was based on eQTL mapping [46]. Rather than using CRISPR-ko, their updated approach utilized CRISPRi (see Section 2.2 below), and the screen was performed using a multiplex approach where each cell was transduced at a high multiplicity of infection (MOI) with multiple gRNAs targeting multiple enhancers; a median of 15–28 gRNAs were delivered per cell, each targeting a different enhancer, followed by CRISPR droplet sequencing (CROP-seq) scRNA-seq transcriptome analysis. Their screen targeted approximately 6000 enhancers, and 664 cis enhancer-gene pairs were identified overall. To validate the screening results, a subset of the functional enhancers was individually tested using both CRISPRi and CRISPR-ko. Their comparison indicated that homozygous deletion lines showed stronger phenotypic effects than those observed in CRISPRi cell lines, showing its efficiency to validate and model specific functional elements. Nevertheless, achieving homozygous deletion is relatively difficult, making establishing homozygous lines a time-consuming process.

CRISPR-ko screens have also been used, with limited success, to identify functional lncRNAs [86]. The Wei group [40] performed a genome-scale deletion screen with a pgRNA library to delete whole sequences of lncRNA genes with known or putative roles in cancer or other diseases. Their screen identified fifty-one lncRNAs affecting cancer cell growth, and nine of these were validated using methods such as functional rescue and gene-expression profiling. While effective, their screen only targeted approximately 700 lncRNAs with over 12,000 sgRNA pairs, making it useful only for small-scale screening. A comparable pgRNA screening strategy was also attempted recently by Tao et al. [41], who screened approximately 600 lncRNAs with a library of nearly 13,000 sgRNA pairs; their approach was similarly difficult to scale up. Subsequently, the Wei group removed the need for pgRNAs by designing a more comprehensive, efficient, and high-throughput screen with a library of sgRNAs that were targeted to either the 5′ splice donor or 3′ splice acceptor sites of lncRNAs to induce exon skipping or intron retention, respectively [42]. This screen targeted over 10,000 lncRNAs to disrupt their function in K562 leukemia cells and was validated using pgRNAs to excise whole exons from a subset of the lncRNAs. However, targeting splice sites is not effective for lncRNAs with only a single exon. Importantly, this study highlighted the importance of taking into consideration the genomic context when designing sgRNAs to target lncRNAs. Since many lncRNAs overlap or are near other genes, targeting lncRNAs with CRISPR-ko may lead to a high rate of false positives [87], and therefore requires additional validation. Interestingly, although the library of Liu et al. targeted antisense lncRNAs overlapping protein-coding genes, their evidence shows that lncRNAs and protein-coding genes do not share the same phenotype following perturbation [88].

Other than its high-throughput applications, CRISPR-ko has been used to more precisely map the functions of individual lncRNAs. Yamazaki et al. [89] analyzed the nuclear enriched abundant transcript 1 (NEAT1) lncRNA to identify which functional domains are important for paraspeckle formation, RNA stability, and isoform switching. To this end, the researchers used pgRNAs to generate 21 sequential deletions (0.6–1.9 kb) tailing NEAT1. Using this approach, the authors identified several functional domains for NEAT1. Similarly, Gaertner et al. [90] found that the lncRNA LINC00261 is important for endocrine development. LINC00261 contains multiple putative ORFs, and, to discriminate their functional role, the authors mutated or deleted each ORF using CRISPR. In this way, the authors determined that the RNA itself was required for endocrine differentiation rather than its produced microproteins.

The DSBs that are generated by Cas9 can be used for introducing foreign DNA into the break site—knock-in; most often by homology directed repair (HDR). This approach has been used to knock in a synthetic polyadenylation signal (spA) proximal to the transcription start site (TSS). It was shown that inserting three constitutive spAs leads to the termination of transcription immediately after the insert [44]. While this approach allows for the control of transcript length and can provide knowledge regarding the different roles of transcription initiation and the transcript itself, generating this model is time-consuming and requires multiple steps.

Overall, CRISPR is useful for the study of non-coding RNAs and genomic regulatory elements, especially when targeting predicted functional sequences. Improving the design of pgRNAs for the deletion of functional elements and for library construction will allow for more robust and efficient ncRNA studies in the future [91,92].

### 2.2. lncRNA Knockdown by CRISPRi

The typical 1–3-nucleotide indel resulting from Cas9 endonuclease activity followed by NHEJ repair is not likely to affect the expression or function of lncRNAs due to their lack of an active ORF. To overcome these limitations, there was a need to adopt CRISPR-based tools that were modified to regulate transcription. While it was shown that modifying sgRNA length can prevent efficient endonuclease activity by Cas9, it can also interact with the DNA and perturb transcription when recruited downstream to the TSS [93]. Thus, the overall efficiency of this approach is limited.

Qi et al. [21] found that a mutated, catalytically dead form of Cas9 (dCas9) could bind a target site to sterically hinder RNA polymerase binding and silence gene transcription without modifying the genome. The same group then found that the addition of a repressive effector protein domain, particularly the Krüppel-associated box (KRAB) DNA binding domain of the KOX1 protein (Figure 3A), improved the knockdown of gene targets significantly compared to dCas9 alone [22,94]. When comparing the relative efficiency of CRISPR-ko, CRISPRi, and the more traditional shRNA in identifying a set of essential protein-coding genes, it was found that CRISPR-ko outperformed CRISPRi and shRNA in terms of both off-target activity and consistency [95]. Importantly, differences in efficiency, accuracy, and off-target effects may affect the phenotype of specific genes. Hence, when comparing large genetic screens that were performed using different perturbation technologies, there is only a moderate overlap between the set of foundational genes in each study [95,96]. Therefore, the choice of technology and library design play an important role in achieving meaningful and reproducible results.

New insights into sgRNA design [97] and improved TSS annotations have successfully improved the efficiency of CRISPRi to be comparable to that of CRISPR-ko [96,98,99]. This point is particularly important for lncRNAs and other non-coding regions because their annotation is not complete, which may affect the efficiency of sgRNA design. Furthermore, it was shown that chromatin accessibility has a major impact on the success of CRISPR knockout and knockdown [100,101,102,103]. In recent years, additional repressive domains have been added to the dCas9-KRAB cassette to further improve the knockdown capabilities of the CRISPRi system. These additional repressive domains were chosen by screening multiple domains from DNA-binding proteins. Notably, the KRAB domain from zinc finger imprinted 3 (ZIM3) (instead of that from KOX1) (Figure 3B) [104], methyl CpG binding protein 2 (MeCP2) (Figure 3C) [105], and SIN3-interacting domain (SID) from MAX dimerization protein 1 (MXD1) (Figure 3D) [86] were found to improve the efficiency of repression by CRISPRi. Importantly, although CRISPRi was designed to create a barrier leading to the collapse of the RNA polymerase complex in a local and transient manner [22], in some cases CRISPRi may also lead to changes in methylation and hence to the silencing of neighboring genes [106]. This may be particularly relevant for antisense and divergent lncRNAs, which are found in close proximity to functional protein-coding genes.

While epigenetic changes may be an undesirable outcome of the dCas9-KRAB fusion, other domains have been added to dCas9 that were specifically designed to generate epigenetic changes. Notably, the fusion of dCas9 to lysine-specific histone demethylase 1 (LSD1) (Figure 3E) was used to target and identify enhancers specifically, whereas dCas9-KRAB targets both enhancers and promoters [107]. The CRISPRi system was also modified to alter DNA methylation states—and thus achieve more long-term repression effects—by fusing dCas9 to the DNA methyltransferase 3A (DNMT3A), DNMT3L (Figure 3F), or ten-eleven translocation 1 (TET1) domains [108,109,110].

The repression effect of KRAB-based CRISPRi is transient and the effect is reduced until elimination at six to fourteen days after transfection [53]. In contrast, epigenetic changes are more stable and can persist after rounds of cell divisions. Constructing a single dCas9 unit that is fused to the KRAB and DNMT3 (A and L) domains showed improved gene silencing over time (Figure 3G); this system was termed CRISPR-off [53]. DNMT3 is known to silence gene expression via the methylation of CpG islands. Nevertheless, CRISPR-off also showed persistent silencing of genes that were lacking annotated canonical CpG islands, showing its efficiency for a broad range of genes. Because methylation is reversible, Nuñez et al. [53] also showed the re-activation of transcription of previously silenced genes by generating dCas9 fused to a domain of the TET enzyme, thereby removing the methylation marker and recruiting transcriptional activator domains to the sgRNA site (see Section 2.3 and Figure 4F below). This CRISPR-off and -on system enables persistent, but reversible, gene silencing using transient expression of the dCas9 cassette.

In recent years, CRISPRi platforms have been adopted for functional screenings of non-coding regulatory DNA elements (e.g., enhancers and promoters) and ncRNAs. For large-scale screening of perturbations, Liu et al. developed pooled and sub-pooled libraries of sgRNAs [47,48]; their pooled library targeted the TSSs of over 16,000 lncRNAs, with 10 sgRNAs per TSS. For the generation of this comprehensive library, the authors used lncRNA annotations from several catalogues and compared them to their expression in seven human cell lines, including induced pluripotent stem cells (iPSCs) and cancer cell lines. Next, the TSSs were annotated and compared to other datasets. Overall, after filtering the data, their library targeted 16,401 lncRNA loci. The authors also divided this library into sub-libraries according to the expression of the target lncRNAs in different cell lines. The sgRNAs were designed according to the hCRISPRi-v2 library, which considers the position relative to the TSS (−25 to +500) and additional features such as chromatin accessibility and nucleotide composition [100]. Using their libraries, the authors then screened for lncRNAs affecting fitness in six human cell lines; they found nearly 500 different lncRNAs significantly affecting cell growth. An important finding of this pioneering screen was that most functional lncRNAs displayed a cell type-specific effect while similar experiments targeting protein-coding genes show that between one-third and half of the identified essential genes are shared between multiple cell types across different origins [111,112].

More recently, Haswell et al. [49] generated a pooled sgRNA CRISPRi library targeting 12,611 lncRNAs that are expressed in human embryonic stem cells (hESCs). The library was designed using the CRISPick tool (formerly Genetic Perturbation Platform) [96,113] based on TSSs predicted by the FANTOM5 CAGE-associated transcriptome (FANTOM-CAT) [114], and contained a total of 111,801 unique sgRNAs (roughly ten sgRNAs per gene). The authors screened for genes affecting hESC differentiation; they identified 60 functional lncRNAs, of which several were functionally validated [49]. However, among the 23 positive control protein-coding genes in the library, only six were identified as positive hits. This finding emphasizes that CRISPRi remains limited in terms of sensitivity, suggesting that the number of functional lncRNAs may be significantly greater than what is currently known.

Importantly, a positive phenotypic correlation was found for matching cell lines and sgRNAs in the studies of Liu et al. and Haswell et al. [47,49], indicating the robustness of this approach. Furthermore, both groups analyzed the correlation between different genetic and genomic features and hit lncRNAs. In both studies, the expression level and distance from the nearest TSS were the most important features that were correlated with functionality. Interestingly, both studies identified enriched lncRNAs at loci overlapping or near cancer-associated SNPs. While both studies identified majorly different features between functional and non-functional lncRNAs, the prediction of functionality based on these features remained low overall. This could be improved by including more features and by improving CRISPRi sensitivity.

In vivo screening is a powerful approach to study the function of genes in the complex microenvironment of an organ. Therefore, Liu et al. [45] performed in vivo CRISPRi screening using a library of 8560 sgRNAs targeting 1503 Wnt-regulated lncRNAs. To achieve high coverage, the authors split this library to three sub-libraries, which were each transduced into FPAD-II cells. Next, the cells were subcutaneously injected into immune-compromised mice. Interestingly, the authors found more lncRNAs affected growth in their in vivo screen compared to those that were identified in matched in vitro screens. Importantly, they observed a low overlap of hits between the two screens which may reflect the context-dependent function of lncRNAs.

One of the most exciting developments in CRISPR screening is the ability to study the function of perturbed genes in an unbiased manner based on transcriptional changes. A major challenge is to find a direct link between the scRNA-seq results and the sgRNA, since sgRNAs lack a poly-A tail. Several approaches were suggested to solve this problem [115,116,117,118,119], and are based on using the sgRNA as a barcode under the transcriptional regulation of RNA polymerase II or by direct tagging of the sgRNA. Optimizing such CRISPRi scRNA screens is expected to further shed light on the function of lncRNAs.

High-throughput CRISPRi approaches are also useful for the high-resolution mapping of regulatory elements in specific regions or surrounding a gene of interest. Fulco et al. [50] used a high-throughput CRISPRi library of 98,000 sgRNAs to map functional lncRNAs that were acting as regulatory elements in a ~1.2 Mb region surrounding the MYC oncogene and the GATA binding protein 1 (GATA1) transcription factor. To screen for regulatory elements, the authors took advantage of the fact that MYC and GATA1 are essential genes in K562 cells. Hence, sgRNAs targeting positively associated regulatory elements will drop out of the cell population over time. Since sgRNA efficiency can vary, the authors calculated the effect of 20 constitutive sgRNAs spanning an average of 314 bp. As expected, the stronger effect was found in sgRNAs targeting the TSS. However, this approach also identified distal regulatory elements, at a distance of up to 1.9 Mb, regulating MYC and GATA1. Besides the TSS of the gene itself, regulatory elements were found overlapping other genes, both coding and lncRNA. For example, sgRNAs targeting the enhancers that were located inside lncRNA PVT1 had a negative effect on cell proliferation in an RNA-independent manner. Surprisingly, Cho et al. found that the PVT1 promoter acts as a tumor suppressor [81], consistent with previous studies showing that targeting the PVT1 TSS leads to increased MYC expression and cell proliferation [47]. These findings suggest an interesting model in which both MYC and the PVT1 promoter compete on the same set of enhancers, and that the activation of the PVT1 TSS prevents the activation of MYC. Nevertheless, these findings highlight the complexity of non-coding regulatory elements and lncRNAs.

A different approach utilized epigenome editing to map functional regulatory elements with a system known as CRISPR-Cas9-based epigenomic regulatory element screening (CERES) that uses both CRISPRi and CRISPRa (see Section 2.3 below) for parallel loss- and gain-of-function screening, respectively [51]. In their CRISPRi screen, the authors used the dCas9-KRAB cassette. The screen was designed to target DNase I hypersensitive sites (DHSs) within a 4.5-Mb region surrounding the β-globin locus in K562 cells with a library of approximately 10,000 gRNAs targeting 281 DHSs within that locus. Next, they screened a 4-Mb region surrounding human epidermal growth factor receptor 2 (HER2), a gene whose amplification is associated with poor prognosis in breast cancer, in A431 epidermoid carcinoma cells. Both screens identified several DHSs whose inhibition modulated the expression of their associated genes. Furthermore, they found that the specific gRNAs with a high functional correlation differed somewhat based on cell type and the direction of perturbation, indicating that these factors are highly relevant for identifying a broad range of functional elements.

Traditionally, reverse genetic screens are based on genetic perturbations and the measurement of distinct cellular phenotypes. Recent advances in CRISPR and scRNA-seq have led to the development of screening approaches based on the measurement of changes in the transcriptome as a proxy for phenotype [115]. Xie et al. [52] developed Mosaic-seq that was based on barcoded CRISPRi (dCas9-KRAB) scRNA-seq screening, to identify functional enhancers at the single cell resolution. The authors generated a pooled library of 51,448 sgRNAs targeting 71 constituent enhancers from 15 super-enhancers. Using this method, they identified several functional enhancers and described the important biological properties of such regulatory elements. Large-scale CRISPR-based screens in conjunction with scRNA-seq is a powerful approach that will enhance our understanding of the non-coding genome.

Overall, CRISPRi is currently the most widely used CRISPR approach for the identification of functional lncRNAs. However, its limited sensitivity remains a barrier to the effectiveness of this strategy [86]. Improvements in gene annotations, CRISPRi approaches, and sgRNA design are expected to shed light on many more functional elements in the non-coding genome.

### 2.3. lncRNA Overexpression by CRISPRa

The high tissue specificity and low expression of lncRNAs make it challenging to study them. Overexpression is a complementary approach to gene knockout and knockdown and is useful for the study of lncRNAs with low expression. To achieve overexpression, exogenous RNA or DNA sequences of the gene of interest can be delivered to the cells. These approaches work well for protein coding genes and some non-coding genes (e.g., miRNA), since the mRNA transcripts do not localize and act near the location of transcription but rather are shuttled to the cytoplasm. However, enhancers regulate the expression of nearby genes. Furthermore, the most well characterized function of lncRNAs is their ability to regulate the expression of neighboring genes (cis regulation) [68]. Therefore, studying lncRNAs and enhancers in their genomic context is critical, especially when inducing gene overexpression. The first dCas9-based transcription activation (CRISPRa) system was a fusion of dCas9 and the multimeric VP64 transcriptional activation domain, comprising four repeats of the viral VP16 domain (Figure 4A) [22,23]. However, this system did not consistently increase target gene expression to sufficient levels and was not sufficiently stable. Therefore, two main approaches were developed to improve CRISPRa: adding more copies of VP16 or adding further TF domains. CRISPR SunTag is based on dCas9 fused to a protein scaffold that can recruit multiple copies of VP64 (Figure 4B) [120]. Balboa et al. [121] took a different approach by further polymerizing the VP16 transactivation domain to form dCas9-VP192 (Figure 4C). On the other hand, to increase transcriptional activation, Konermann et al. [24] combined the dCas9-VP64 fusion protein with two additional transcriptional domains—p65 and the heat shock factor 1 (HSF1)—as well as the MS2 binding protein to recruit the domains to a modified sgRNA to generate the synergistic activation mediator (SAM) system (Figure 4D). Chavez et al. [122] fused dCas9 to a triple activator fusion, VP64-p65-Rta, to create the VPR system (Figure 4E). Subsequently, a hybrid of the SunTag protein scaffold and p65-HSF1 from the SAM approach was created to generate the SunTag-p65-HSF1 (SPH) cassette [123]. Interestingly, SPH showed superior gene activation not only in vitro, but also in transgenic mice.

CRISPRa has also been adapted for gene activation via epigenome editing. By fusing dCas9 to the catalytic core domain of acetyltransferase p300 (Figure 4G), Hilton et al. [124] designed a CRISPRa system that targets promoters and both distal and proximal enhancers to induce histone H3K27 acetylation for the transactivation of the gene of interest. An advantage of the epigenome editing approach is that transient transfection of the sgRNA can be used to induce long-term transcriptional activation of the target [124].

This dCas9-p300 cassette was used by Klann et al. [51] as the CRISPRa component of their CERES system (see Section 2.2 above). Using the same gRNA library targeting the 4-Mb region surrounding HER2, the authors performed a CRISPRa screen in HEK293, a cell line with low endogenous HER2 expression. Their CRISPRa screen was consistent with and mirrored the CRISPRi screen with corresponding gRNAs generally showing similar effects in the opposite direction [51]. As mentioned above, their results also highlighted the benefits of performing both gain- and loss-of-function screens to identify a wider range of functional elements.

Li et al. [125] designed the enCRISPRa system based on a dCas9-VP64 or -P300 fusion protein, as well as the recruitment of core P300 or VP64 to MS2 hairpin motifs added to the sgRNA. enCRISPRa showed superior activation of the target genes when dCas9 was fused to P300 and VP64 was recruited to the sgRNA (Figure 4H). The authors compared the efficiency of their approach to that of previously established systems by targeting enhancers and measured the transcriptional activation of their target genes. Interestingly, for the studied enhancers, enCRISPRa was more potent than the VPR, SunTag, and SAM approaches [125].

While the SAM approach and enCRISPRa use MS2 hairpins and MS2 coat protein (MCP) to recruit additional activators to the dCas9-sgRNA complex, it was shown that the MS2-P65-HSF1 fusion has a low lentiviral titer. To improve the efficiency of this approach, Sanson et al. [96] replaced MS2 with a PP7 binding site and the PP7 bacteriophage coat protein (PCP) heptamer. This improvement may be especially relevant for primary cells and in vivo studies where a high MOI is critical and limiting [126].

Several high-throughput CRISPRa screens have been performed targeting enhancers and lncRNAs with various aims.

In 2017, Simeonov et al. [58] used a CRISPRa tiling approach to identify enhancers that were regulating the expression of the autoimmune-related interleukin-2 (IL-2) receptor A (IL2RA). The authors used dCas9-VP64 and a library of 20,412 gRNAs tiling 178 kb around the IL2RA locus. IL2RA is a cell surface receptor; hence, it can be easily identified by antibody staining and sorted based on its expression level. In this way, the authors identified six enhancers that stimulate IL2RA expression.

Joung et al. [57] investigated lncRNAs with a role in conferring resistance to the BRAF inhibitor vemurafenib in melanoma cells by targeting over 10,000 lncRNAs. Their screen identified 11 lncRNA loci whose activation enhances vemurafenib resistance. Of those, at least one acts by inducing the expression of four neighboring genes, one of which is associated with the resistance phenotype. This approach is applicable not only for investigating vemurafenib resistance, but also can be applied to various treatments for many different cancers. Koirala et al. [56] used SAM and an AKT reporter to screen for lncRNAs that were associated with AKT activity, a major member of the critical phosphoinositide 3-kinase (PI3K) cell signaling pathway. After screening a specific group of lncRNAs, AK023948 was identified as a positive regulator of AKT. The authors determined that AK023948 functionally interacts with DExH-Box Helicase 9 (DHX9) and p85 to modulate AKT activity. Similarly, Bester et al. [55] designed a comprehensive and integrated CRISPRa library to screen for both coding and non-coding genes that are associated with resistance to the acute myeloid leukemia treatment cytarabine. Their approach differed in that it focused on identifying interactions between protein coding gene and lncRNA pairs, thus establishing a broad characterization of mechanisms leading to chemotherapy resistance. They found that the upregulation of the growth arrest specific 6-antisense 2 (GAS6-AS2) lncRNA leads to the hyperactivation of the GAS6/TAM pathway, which is an important resistance mechanism in multiple cancer types.

Interestingly, large-scale studies of CRISPRi and CRISPRa found only a moderate overlap between hits that were derived from the reciprocal approaches [51,127,128]. This may be due to differences in the efficiency of transcription regulation or owing to the biological effects of the target gene.

Overall, CRISPRa is a useful approach to identify and study cis and trans regulation. Therefore, it is especially relevant for studying non-coding regulatory genes and DNA elements.

### 2.4. Novel Alternative Tools: Cas12 and Cas14

Additional Cas enzyme families are also being investigated as alternative or complementary systems to the established Cas9 family. Recently, Cas12a (previously Cpf1) was discovered as a distinct enzyme family for genome editing similar to Cas9 [29]. Several key differences distinguish Cas12a from Cas9: the immature crRNA is processed directly into a gRNA by Cas12a and does not require a tracrRNA; Cas12a cleaves the DNA upon recognition of a T-rich PAM sequence, in contrast to the G-rich PAM recognized by Cas9; cleavage is distal to the PAM sequence; and cleavage produces a 4–5-nt 5′ overhang [29,129]. Several other Cas12 families were discovered recently, in particular Cas12b [130]; several others have also been characterized, including Cas12c, -g, -h, and -i [131].

A major advantage of the recently discovered Cas14 [30] over Cas9 and other Cas enzymes is its small size; Cas14 variants range from 400–700 amino acids, in contrast to the 950–1400 amino acid length of Cas9 and other previously characterized Cas proteins [30]. Cas14 shows targeted, non-specific ssDNA cleavage. Interestingly, the enzyme does not require a PAM for activation, but is highly sensitive to mismatches in the middle of the gRNA target region. This makes Cas14 potentially useful for diagnostic purposes as it can be used to accurately identify ssDNA pathogens or to detect SNPs at a high fidelity without the constraint of a PAM [30]. Research into the novel Cas14 system is minimal at present and future discoveries may yield strategies to induce PAM-directed dsDNA cleavage by Cas14, allowing it to be used as a screening tool as well as a diagnostic one.

A potential advantage of using different Cas homologues is the ability to potentially target two different genes simultaneously with minimal risk of overlapping effects—such as the simultaneous overexpression of one gene with CRISPRa and knockdown/knockout of another using Cas12.

### 2.5. RNA Targeting by Cas13

Cas9-based technologies target the DNA to introduce mutations, interfere with transcription, or introduce epigenetic modifications; however, recently it was found that programmable CRISPR can go beyond targeting double-stranded DNA. Cas13 is a programmable ribonuclease that can bind single-stranded RNA targeted by the crRNA [132]. While several orthologs of Cas13 have been identified, Cas13d was found to be the most effective for gene knockdown in mammalian cells [133]. Cas13 uses a ~54-nt gRNA, and target specificity is encoded by a 23–30-nt spacer that is complementary to the target region. Unlike Cas9, Cas13 does not require a protospacer flanking sequence when targeting RNAs in mammalian cells. However, sgRNA efficacy varies dramatically based on the gRNA and target features [32,59,60]. The sgRNA is sensitive to mismatches, especially in positions 15–21. Furthermore, although single or double mismatches are tolerated when they do not overlap the critical sequence, triple mismatches have a detrimental effect on knockdown efficiency. Other features of the sgRNA, such as nucleotide composition and sgRNA self-folding, also affect the efficiency. Importantly, Cas13 shows a strong preference for single-stranded RNA while RNA transcripts tend to form secondary structures, generating regions of double-stranded RNA that are resistant to targeting by Cas13 [134]. Hence, sgRNAs may require further optimization based on the target sequence to achieve the maximum effect. Furthermore, Cas13 can be designed with a nuclear localization signal (NLS) or as a cytoplasmic protein with a nuclear export signal (NES). The NES appears to be largely unnecessary as Cas13-NLS was found to stimulate a strong reduction even when targeting mRNA transcripts that were translated in the cytoplasm [59].

While Cas13 is a useful and complementary approach to target mRNA and lncRNA transcripts [61], one important use of Cas13 is its ability to target circular RNAs (circRNAs), which are stable and functional products of aberrant RNA splicing. Specific targeting of circRNAs is dependent on the unique sequence of the back-splicing junction (BSJ) that creates the circRNA. This BSJ site is the only sequence that is unique to the circRNA and does not appear in the linear RNA. Both siRNAs and Cas13 can target circRNAs, although Cas13 outperformed siRNA in terms of both efficiency and accuracy [60,62].

Visualizing the location and dynamics of DNA and RNA is critical for understanding their function. Classical approaches for RNA labeling, such as fluorescence in situ hybridization (FISH), require a set of unique oligonucleotide-based probes that are labeled with fluorophores, making these approaches expensive and time-consuming. However, the modularity of CRISPR systems raises the possibility of developing a flexible system for nucleic acid visualization [135,136,137]. Using two dCas13b variants with a fluorescent tag, Yang et al. [138] were able to label and track both lncRNA and mRNA molecules in live cells. This approach showed marked improvements over previously established live-cell imaging techniques, including a reduced signal-to-noise ratio. Furthermore, the addition of an NLS to the probe improved the labeling of nuclear transcripts without affecting the efficiency of cytoplasmic element labeling [138]. One important advantage of the CRISPR system for labeling and visualization is the ability to differentially label the DNA by dCas9 and the RNA by dCas13 at the same time; in this way it is possible to study the relationship between DNA dynamics and RNA transcription.

Studying protein-RNA interactions is critical to understanding their biological processes. Many lncRNAs were found to interact with proteins and are important for their localization and function (e.g., XIST, NEAT1, and NORAD). To study this interaction, RNA pulldown is often used to identify RNA binding proteins in an unbiased manner. However, RNA pulldown requires synthetic RNA or oligonucleotide probes. As an alternative approach, several groups have developed CRISPR-based systems to study RNA-protein interactions. These methods are based on dCas13 fused to an enzyme-catalyzed proximity labeling probe [139,140]. Once the sgRNA-dCas13 targets RNA, the labeling enzymes biotinylate nearby proteins that can be later pulled down based on the biotin label and identified using mass spectrometry. In the future, these approaches may simplify and advance our understanding of RNA-protein interactions, as well as their functional roles.

Another approach using dCas13 as an RNA-binding platform is the adaptation of dCas13 for precise transcript editing by constructing a fusion protein of the dCas13 and a base editing enzyme such as adenosine deaminase acting on RNA type 2 (ADAR2) [141,142]. Once targeted to a specific site, ADAR can edit adenosine to inosine (A-to-I), which is identified by the ribosome as guanine (G). Additional modifications to ADAR2 enable cytidine-to-uridine (C-to-U) editing [142]. This approach was previously developed using the oligonucleotide-recruiting ADAR2 protein. However, adapting this system with other RNA editors may open new options for correcting disease-associated mutations without the need for irreversible DNA mutagenesis. This is also an important tool to study lncRNAs since much of the A–I editing occurs in non-coding transcripts.

Overall, RNA editing is an important complementary approach to study lncRNAs, especially when trying to differentiate between DNA- and RNA-dependent functions of lncRNAs.

## 3. Approaches Other Than CRISPR

There are several other strategies that have been used as alternatives to CRISPR-Cas9 to perturb non-coding gene expression. The major classical alternatives to CRISPRi are RNAi, using siRNA or shRNA; and ASOs, usually via locked nucleic acids (LNAs). These techniques are useful as complementary approaches for CRISPR-based perturbation experiments.

RNAi is a major mechanism of post-transcriptional gene silencing in animals and plants; dsRNA sequences that are complementary to the target mRNA bind the transcript to target it for degradation. RNAi was originally discovered in *C. elegans*, and was first used for targeted gene silencing in mammalian cells by Elbashir et al. [143], who designed 21-nt siRNA duplexes to suppress target genes. Since then, RNAi has become a popular method for targeted and high-throughput gene silencing [144,145,146]. Researchers have attempted high throughput siRNA screens to identify novel lncRNAs with limited success. Nötzold et al. [33] performed a screen targeting 638 lncRNAs deregulated in cancer to determine their impact on cell growth and morphology in HeLa cells; the authors used time-lapse microscopy to monitor changes in morphology and cell cycle progression. In this way, 26 putative lncRNAs were identified; of them, LINC00152 was expressed in a wide range of cell lines and was confirmed as a major factor that was required for mitosis, as its knockdown led to prometaphase arrest. Stojic et al. [147] performed a larger screen of 2231 lncRNAs with an siRNA library. They identified two lncRNAs affecting the progression of mitosis in HeLa cells: LINC00899 and C1q & TNF-related 1-antisense 1 (C1QTNF1-AS1). While the scale of this more recent screen was higher than that of Nötzold et al., both studies were performed at a scale much lower than that of similar CRISPRi/a screens that were discussed above. As with all gene perturbation strategies, the off-target effects of siRNA must be considered to correctly interpret the observed phenotypic effects. Jackson & Linsley [148] reviewed the different types of off-target effects that were caused by siRNA that must be accounted for to improve their accuracy in genetic screens and to promote their therapeutic potential. Major off-target effects include miRNA-like regulation of unintended transcripts with partial sequence complementarity, which can be reduced by introducing redundancy or by pooled siRNAs; stimulation of the immune response by the siRNA or its delivery vehicles, which can be minimized through chemical modifications; and saturation of the endogenous RNAi machinery leading to widespread disruption of miRNA processing and function.

LNAs are nucleic acid analogues with a 2′-O,4′-C-methylene modification that have a high affinity for RNA. They were originally developed in 1998 [149], and were first used to study nuclear lncRNAs by Sarma et al. [150], who targeted the XIST RNA with ASOs to determine its sequence requirements and the kinetics of its localization to the X chromosome. An overall protocol for LNA-based knockdown of regulatory and other non-coding RNAs was recently described by Roux et al. [85]. ASOs have also been tested in vivo for potential therapeutic use; Wheeler et al. [151] used ASOs in a transgenic mouse model of the degenerative disease dystrophia myotonica type 1 (DM1), characterized by expanded CUG repeats in the 3′ UTR of DM1 protein kinase (DMPK). They found that the expanded repeats allele was highly sensitive to targeting by ASOs; treated mice displayed a marked knockdown of the mutant allele along with a reversal in the pathological, histological, and transcriptomic features of DM1 for up to a year following systemic treatment with ASOs. Subsequently, ASOs have been used for relatively high throughput lncRNA screens. As a part of the FANTOM6 project, Ramilowski et al. [34] used LNA-modified GapmeR ASOs to knock down 285 lncRNAs in primary human dermal fibroblasts with a total of 2055 ASOs. After confirming successful knockdown, the authors assessed the effect of knockdown on cell proliferation and various morphological features and found that ~30% of the lncRNAs that were targeted affected the growth or morphology of the cells. They also used several techniques for molecular phenotyping—including sequencing 970 Capped Analysis of Gene Expression (CAGE) libraries, Motif Activity Response Analysis (MARA), and Gene Set Enrichment Analysis (GSEA)—to validate the observed phenotypes on the transcriptome level. Finally, the authors validated their results by targeting a subset of nine lncRNAs with siRNA, but only three targets showed consistent results. These results highlight one major limitation of ASO technology for high-throughput analyses. The prediction of ASO target knockdown efficiency is generally poor, and, therefore, several ASOs must be designed per target. Because ASOs are more difficult and expensive to design and synthesize than the gRNAs that are used for CRISPRi, fewer genes can be investigated in a single study. The degree of off-target effects that are associated with ASOs was also described by Kamola et al. [152], who found a relatively high rate of off-target effects.

All of these methods, including CRISPRi, have off-target effects that must be accounted for when analyzing results to confirm their accuracy. To this end, several studies have investigated and compared the relative efficiency and off-target effects of each technique. Lennox & Behlke [35] compared the relative knockdown efficiency of RNAi and ASOs against nuclear and cytoplasmic lncRNAs. They found that RNAi was more effective for suppressing cytoplasmic RNAs (e.g., differentiation antagonizing non-protein coding RNA [DANCR], Opa interacting protein 5 [OIP5]-AS1), ASOs were more effective against nuclear RNAs (e.g., MALAT1, NEAT1), and both methods were effective for dual-localized lncRNAs (e.g., taurine up-regulated 1 [TUG1], HOX antisense intergenic RNA [HOTAIR]). This indicates that the subcellular localization of the lncRNA must be considered when choosing between these two methods. Subsequently, Stojic et al. [36] characterized the off-target effects of CRISPRi, ASOs, and siRNA for loss-of-function transcriptional analyses. Their analysis found that each method has distinct off-target effects. RNAi and ASOs both caused differentially expressed genes (DEGs) as a result of treatment with the transfection reagent only (~30 DEGs) and following treatment with non-targeting controls (~50–100 DEGs). The DEGs were not associated with any single functional pathway, indicating that these off-target effects are difficult to predict or remove computationally. Additionally, each non-targeting negative control perturbed a distinct set of genes. The authors also found that CRISPRi using dCas9-KRAB led to transcriptional variations in clonal cells after transduction with dCas9-KRAB (201 DEGs) and following the transduction of non-targeting gRNAs (~100 DEGs). However, these effects were minimal in non-clonal cells following the same treatments (3 and 8 DEGs, respectively). Only two genes were differentially expressed between the two gRNAs, indicating that CRISPRi is highly specific compared to RNAi and ASOs. Finally, in a comparative analysis targeting a specific non-characterized lncRNA (solute carrier family 25 member 25 [SLC25A25]-AS1) using all three methods, the authors found no common DEGs other than SLC25A25-AS1 itself; notably, fewer DEGs were found for CRISPRi than for ASOs. Importantly, knockdown by ASOs resulted in mitotic delay, while this phenotype was not observed using CRISPRi. Overall, the most major discrepancy that was observed in their study was that between different knockdown strategies [36].

## 4. Conclusions

Recent developments in gene editing approaches open new possibilities for studying lncRNAs and other non-coding genes and genomic elements. Since 2013, the field of genetic engineering and CRISPR-based approaches has developed rapidly. These developments have opened new possibilities and generated many tools for studying the non-coding genome. Currently, CRISPR-based technologies give researchers different options to manipulate the expression of lncRNAs by either decreasing or increasing the expression of genes of interest. Other approaches can help distinguish RNA function from that of DNA regulatory elements; for example, by identifying overlapping enhancers or by directly targeting the RNA molecules. Additional CRISPR technologies are currently being developed to enable the visualization and study of the interactions of lncRNAs with its partners. Notably, not all technologies are fully evolved—in some cases, the technology is biased; in others, the technology may not be sensitive enough to detect lncRNAs with low expression. However, the use of established and newly developing CRISPR technologies will lead to a new understanding of genome function and organization, as well as the discovery of new clinically relevant genes.

## Figures and Tables

**Figure 1 ncrna-07-00079-f001:**
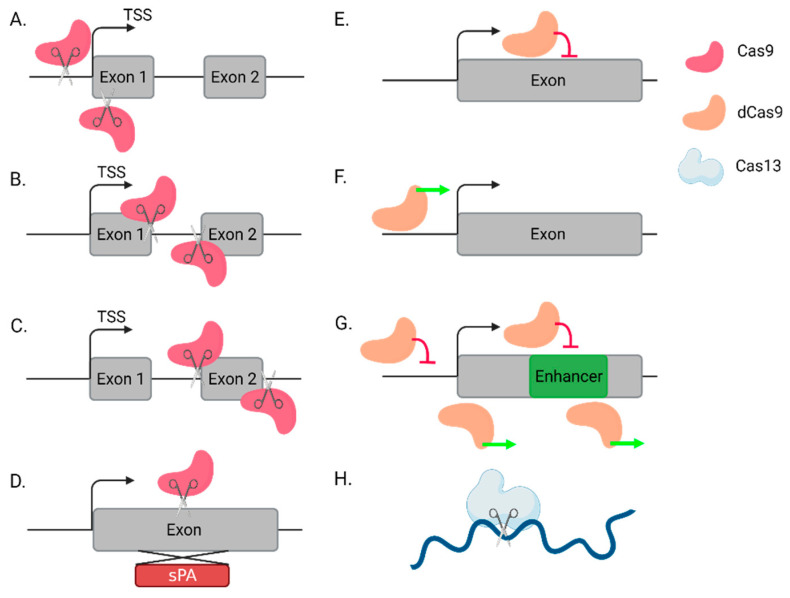
A schematic illustration of CRISPR-based approaches to study lncRNAs. (**A**) Transcription start site (TSS) deletion. (**B**) Mutations of splice sites. (**C**) Removal of an exon or a large genomic fragment. (**D**) Knock-in: insertion of a synthetic polyadenylation (spA) signal. (**E**) Knockdown by CRISPRi. (**F**) Gene overexpression by CRISPRa. (**G**) Tiling a genomic locus using CRISPR-ko/CRISPRi/CRISPRa to identify enhancers, which in many cases can overlap or affect lncRNAs. (**H**) Targeting an RNA transcript by Cas13. (Created with BioRender.com (accessed on 12 December 2021)).

**Figure 2 ncrna-07-00079-f002:**
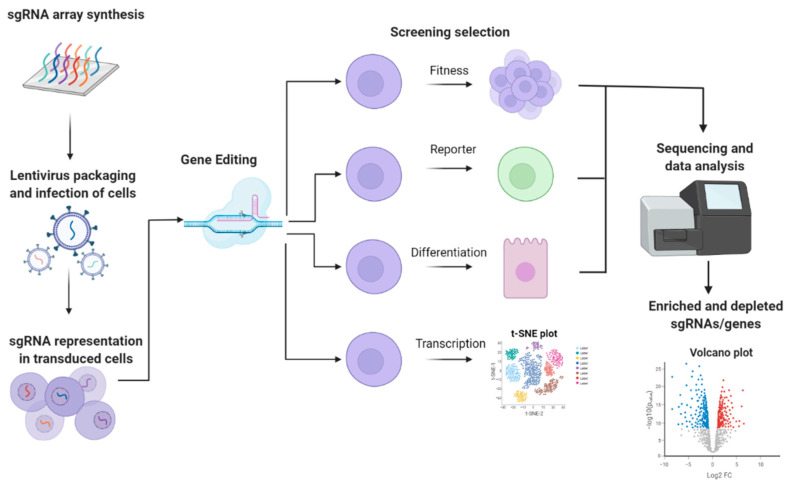
The general approach for high-throughput reverse genetics screening. (Created with BioRender.com (accessed on 12 December 2021)).

**Figure 3 ncrna-07-00079-f003:**
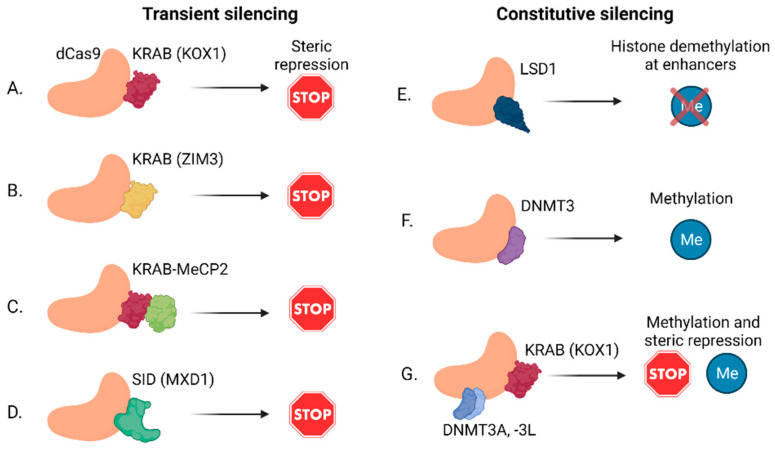
Different approaches for efficient transient (**A**–**D**) or constitutive (**E**–**G**) gene silencing using CRISPRi. (**A**) dCas9 fused to the KRAB domain of KOX1. (**B**) dCas9 fused to the KRAB domain of ZIM3. (**C**) dCas9 fused to KOX1 KRAB and MeCP2. (**D**) dCas9 fused to the SID domain of MXD1. (**E**) dCas9 fused to DNMT3 leading to DNA methylation at the target site. (**F**) dCas9 fused to LSD1 leading to histone demethylation at enhancer regions. (**G**) CRISPR-off, comprising dCas9 fused to KOX1 KRAB and DNMT3A and -3L. (Created with BioRender.com (accessed on 12 December 2021)).

**Figure 4 ncrna-07-00079-f004:**
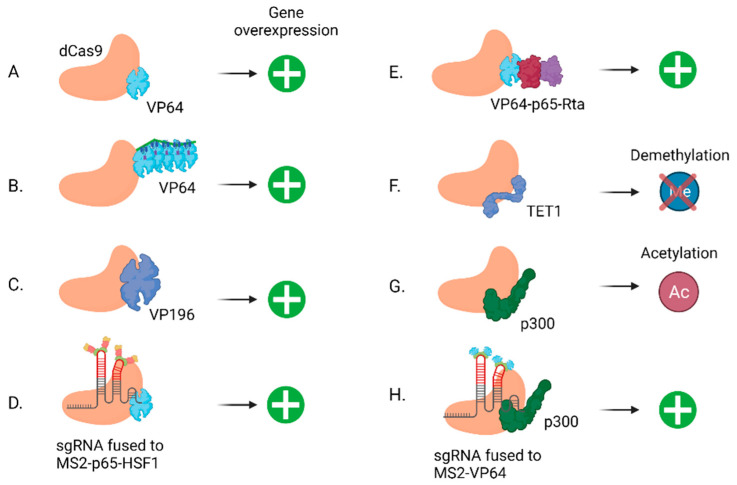
Different approaches for efficient gene activation using CRISPRa. (**A**) dCas9 fused to the VP64 viral repeat domain. (**B**) CRISPR SunTag, comprising multiple copies of VP64 fused to a protein scaffold and recruited to dCas9. (**C**) dCas9 fused to the VP196 viral repeat domain containing more repeats than VP64. (**D**) Synergistic activation mediator (SAM), comprising dCas9 fused to VP64 and a modified sgRNA fused to MS2, p65, and HSF1. (**E**) VPR, comprising dCas9 fused to VP64, p65, and Rta. (**F**) CRISPR-on, comprising dCas9 fused to TET1 to reverse the repressive methylation effect caused by CRISPR-off (See Section 2.2 and Figure 3G above). (**G**) dCas9 fused to p300 leading to H3K27 acetylation for gene transactivation. (**H**) enCRISPRa, comprising dCas9 fused to p300 and a modified sgRNA fused to MS2 and VP64. (Created with BioRender.com (accessed on 12 December 2021)).

**Table 1 ncrna-07-00079-t001:** Major approaches for high-throughput analysis of regulatory elements.

Perturbation Method	Effect	Element(s) Targeted	Reference(s)
CRISPR-ko	Mutagenesis	Enhancers, protein-coding genes (PCG)	[37,38,39]
CRISPR-ko	Transcription start site/whole sequence deletion	lncRNAs	[40,41]
CRISPR-ko	Splice site mutation	lncRNAs	[42]
CRISPR-ko	Tiling deletion	lncRNAs	[43]
CRISPR-ki	Synthetic polyadenylation signal (spA) insertion	lncRNAs	[44]
CRISPRi	Transcription inhibition	lncRNAs, enhancers, PCG	[45,46,47,48,49,50,51,52]
CRISPRi	Epigenetic silencing	lncRNAs, enhancers, PCG	[53]
CRISPRi	Binding site interactions	Enhancers	[50,51,52,54]
CRISPRa	Transcription activation	lncRNAs, enhancers, PCG	[51,55,56,57,58]
CRISPRa	Epigenetic activation	lncRNAs, enhancers, PCG	[53]
Cas13	RNA targeting	lncRNAs, circRNA, PCG	[59,60,61,62]

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
