# Peer review of "CRISPR-Based Approaches for the High-Throughput Characterization of Long Non-Coding RNAs"

_ncrna, 2021, doi:10.3390/ncrna7040079_

Round 1
Reviewer 1 Report
Hazan et al., have demonstrated the different methods targeting lncRNA level and their fundamental roles in biology research. Initially they described the general functions of lncRNAs in cellular processing and discussed the applications of high-through sequencing by silencing and activating lncRNAs with several approaches particularly with CRISPR-CAS9 or CAS13 techniques. They cited a lot number of references to illustrate the advantage and deficiency of these technologies and concluded that the developing CRISPR methodology will lead to broad insight to understanding the functions of human genome, as well as benefit the clinical therapy in near future. It is an interesting review, however, more figures are desired, if possible, which may help audiences to understand the manuscript.
Author Response
“Hazan et al., have demonstrated the different methods targeting lncRNA level and their fundamental roles in biology research. Initially they described the general functions of lncRNAs in cellular processing and discussed the applications of high-through sequencing by silencing and activating lncRNAs with several approaches particularly with CRISPR-CAS9 or CAS13 techniques. They cited a lot number of references to illustrate the advantage and deficiency of these technologies and concluded that the developing CRISPR methodology will lead to broad insight to understanding the functions of human genome, as well as benefit the clinical therapy in near future. It is an interesting review, however, more figures are desired, if possible, which may help audiences to understand the manuscript.”
We thank the reviewer for their helpful comments. Per your suggestion, we have added three new figures to the paper: Figure 2 outlining the overall process of high-throughput reverse genetics screens, Figure 3 outlining the major CRISPRi cassettes, and Figure 4 outlining the major CRISPRa cassettes.
Reviewer 2 Report
The review article submitted by Joshua Hazan and Assaf C. Bester entitled “CRISPR-based approaches for the high-throughput characterization of long non-coding RNAs” provides a good overview of the techniques, with in depth analysis, drawbacks and possibilities of different CRISPR-based approaches. The review is well written and interesting to read. I only have minor issues regarding the manuscript:
- Section 2, page 3 - please specify the reason behind selecting the four lncRNA that are described in section 2 (Proposed and proven functions of lncRNAs)?
- Page 1, line 32-34 – the authors wrote what NORAD standed for, but not XIST, MALAT1 and PVT1. Please provide also names for those lncRNAs.
- The sections could be better organized with more informative titles of each section (for example lncRNA knockdown by CRISPRi instead of CRISPRi). Perhaps a CRISPR-based approaches to study lncRNAs section could be made with subsections, such as lncRNA knockdown by CRISPRi, gene overexpression by CRISPRa (similar to Figure 1).
Author Response
The review article submitted by Joshua Hazan and Assaf C. Bester entitled “CRISPR-based approaches for the high-throughput characterization of long non-coding RNAs” provides a good overview of the techniques, with in depth analysis, drawbacks and possibilities of different CRISPR-based approaches. The review is well written and interesting to read. I only have minor issues regarding the manuscript:
- Section 2, page 3 - please specify the reason behind selecting the four lncRNA that are described in section 2 (Proposed and proven functions of lncRNAs)?”
Thank you for your helpful suggestion. We chose those four lncRNAs as an illustrative example of lncRNAs with a major function in normal cellular processes as well as disease states. We have added the following sentence (section 1.1, line 96) to address this: “Here, we briefly explain four of the most comprehensively characterized and well known lncRNAs with a major function.”
Reviewer 3 Report
The current review article entitled "CRISPR-based approaches for the high-throughput characterization of lncRNAs" intended to summerize the use of different CRISPR/Cas9-based methods for genetic and functional characterization of lncRNAs. But, the authors described and reviewed everything except lncRNAs. The reviewer lost reading the content and disconnected at most of the parts/sections and authors did beat around bush throughput. Very little knowledge to the ncRNA field from this review and authors haven't structured their review well. Though entire review is full of flaws and out of their scope, the reviewer tried to put through some points to convey authors the reasons for disappointment.
1) Firstly, authors mentioned in the abstract and Intro that very handful of lncRNA were characterized due to their low expression. But, the most of techniques (CRISPRi /CRISPRa) described are based on the alteration of lncRNAs expression. There is no discussion how do the researchers overcome to characterize lncRNAs which is suffering due to poor expression.
2) LncRNAs regulates the genomic elements, but are not regulatory elements. The Introduction starts with 'regulatory elements in the genome, and of these lncRNAs...' is misleading.
3) In the abstract, line 18 authors indicated that the reviewed approaches allow for the 'detection' of lncRNAs and authors failed to explore this aspect how CRISPR-based approaches used for the detection of NOVEL lncRNAs in the main content.
4) In the same line, CRISPR-based approaches only allows to KO/KD/Over-express already known lncRNAs and it won't facilitate new lncRNAs identifications other than NGS-based approaches. These appraoches all need sequences to design custom gRNAs.
5) In the figure-1, how 1g (identifying enhancers) will help to identify lncRNAs or characterize the functions of lncRNAs other than enhancers.
6) Section-3 has discussion nothing related to lncRNAs rather than general reverse genetic screening.
7) In every section, some introduction to describe technique is required. But, in this review authors gave more emphasis to describe general CRIPSR applications for miRNAs/PCG/enhancers/rather than important LncRNAs.
8) Please provide summarized screens in table format in which caners/cells to make more interaction to the readers.
9) Authors can use some of the CRISPR screen databases such as http://genomecrispr.dkfz.de/ and https://www.kobic.re.kr/icsdb/ and many more to review the CRISPR screens
10) More importantly, these CRISPR-based approaches will be good for phenotypic screen and lncRNA that exhibit phenotype upon CRIPR-ko/i/a. There are very few lncRNA that have phenotypic impact upon perturbation. So, authors should emphasize more on the techniques such OMICS-based to understand their functional consequences such as changes in the whole transcriptome/chromatin structure/epigenetic alteration/transcription machinery recruitment are critical for genetic and functional characterization.
So, overall the objectives authors intended to meet in the review were not fulfilled.
Best,
Author Response
“The current review article entitled "CRISPR-based approaches for the high-throughput characterization of lncRNAs" intended to summerize the use of different CRISPR/Cas9-based methods for genetic and functional characterization of lncRNAs. But, the authors described and reviewed everything except lncRNAs. The reviewer lost reading the content and disconnected at most of the parts/sections and authors did beat around bush throughput. Very little knowledge to the ncRNA field from this review and authors haven't structured their review well. Though entire review is full of flaws and out of their scope, the reviewer tried to put through some points to convey authors the reasons for disappointment.
1) Firstly, authors mentioned in the abstract and Intro that very handful of lncRNA were characterized due to their low expression. But, the most of techniques (CRISPRi /CRISPRa) described are based on the alteration of lncRNAs expression. There is no discussion how do the researchers overcome to characterize lncRNAs which is suffering due to poor expression.”
We thank you for your comments. We agree with the reviewer that this point needed to be better clarified. CRISPRa is useful for studying lncRNAs with low expression by overexpressing them. Therefore, we added to the CRISPRa section the following sentence (section 2.3, line 467-469): “The high tissue specificity and low expression of lncRNAs make it challenging to study them. Overexpression is a complementary approach to gene knockout and knockdown, and is useful for the study of lncRNAs with low expression.”
“2) LncRNAs regulates the genomic elements, but are not regulatory elements. The Introduction starts with 'regulatory elements in the genome, and of these lncRNAs...' is misleading.”
We have changed this as follows to reflect your comment (line 25-29): “A large proportion of the human genome encodes various types of regulatory elements and non-coding genes. Of these, the long non-coding RNA (lncRNA), defined as a transcript longer than 200 nt with low or no protein-coding potential, is one of the largest—and arguably among the most poorly characterized—classes of non-coding RNA genes.”
“3) In the abstract, line 18 authors indicated that the reviewed approaches allow for the 'detection' of lncRNAs and authors failed to explore this aspect how CRISPR-based approaches used for the detection of NOVEL lncRNAs in the main content.”
Our review focuses on functionally characterizing lncRNAs that have already been discovered, rather than the detection of novel lncRNAs. We have changed the abstract as follows to indicate this (line 18): “These approaches allow for the knock-down, detection, and characterization of lncRNAs. Here, we review how various CRISPR-based strategies have been used to discover characterize lncRNAs with important functions.” We have also modified the introduction as follows (line 38-41): “Very few of the abundant lncRNAs in the human genome have been properly functionally classified and it remains difficult to accurately predict functional lncRNAs computationally; therefore, high-throughput assays are important for identifying and characterizing functional lncRNAs in different tissues and contexts.” We have also modified the abstract as follows (line 11-12): “A major limitation for identifying functional determining the function of lncRNAs was the lack of methodologies suitable for studying these genes.”
“4) In the same line, CRISPR-based approaches only allows to KO/KD/Over-express already known lncRNAs and it won't facilitate new lncRNAs identifications other than NGS-based approaches. These appraoches all need sequences to design custom gRNAs.”
As you have suggested, we have changed the abstract to more clearly indicate that CRISPR-based approaches are useful for characterizing the function of lncRNAs, rather than for discovering new ones. In addition to what we have outlined above, we have made the following changes to address this. Section 1, line 31: “While the majority of lncRNAs identified until now remain uncharacterized…” Section 1, line 61: “These methods, in addition to classical CRISPR-ko, have been adapted in a wide range of strategies and applied for various high-throughput screens of functional lncRNAs in many different cell types to improve the annotation and functional characterization of lncRNAs.”
“5) In the figure-1, how 1g (identifying enhancers) will help to identify lncRNAs or characterize the functions of lncRNAs other than enhancers.”
Enhancers often overlap or directly affect the expression of lncRNAs, and therefore are highly relevant to studies of lncRNAs. We have modified the legend of figure 1g to address this as follows: “(g) Tiling a genomic locus using CRISPR-ko/CRISPRi/CRISPRa to identify enhancers, which in many cases can overlap or affect lncRNAs.”
“6) Section-3 has discussion nothing related to lncRNAs rather than general reverse genetic screening.”
We have added the following to that section to more clearly link reverse genetic screening to lncRNA studies. Line 145-147: “Because thousands of lncRNAs have been identified and only a fraction of them have been characterized, there is a need for robust methods for the identification of functional lncRNAs in different biological contexts.” Line 163: “pooled screening can be performed with up to hundreds of thousands of different manipulations in one experiment and is therefore highly relevant for functional lncRNA screens.”
“7) In every section, some introduction to describe technique is required. But, in this review authors gave more emphasis to describe general CRIPSR applications for miRNAs/PCG/enhancers/rather than important LncRNAs.”
Thank you for this important suggestion. We have removed from section 2.1 (“Using CRISPR-ko to directly or indirectly knock out lncRNAs”) the paragraph discussing miRNAs in order to focus more directly on lncRNAs. We have kept the sections on enhancers, as they often overlap or regulate lncRNAs and are therefore closely connected to lncRNAs. Further, we have tried to mention protein-coding genes only where they are directly related to lncRNAs or the screening method being described.
“8) Please provide summarized screens in table format in which caners/cells to make more interaction to the readers.”
We thank the reviewer for this comment. Please see table 1.
“9) Authors can use some of the CRISPR screen databases such as http://genomecrispr.dkfz.de/ and https://www.kobic.re.kr/icsdb/ and many more to review the CRISPR screens.”
We thank the reviewer for this comment. Please see table 1.
“10) More importantly, these CRISPR-based approaches will be good for phenotypic screen and lncRNA that exhibit phenotype upon CRIPR-ko/i/a. There are very few lncRNA that have phenotypic impact upon perturbation. So, authors should emphasize more on the techniques such OMICS-based to understand their functional consequences such as changes in the whole transcriptome/chromatin structure/epigenetic alteration/transcription machinery recruitment are critical for genetic and functional characterization.”
Coupling CRISPR-based approaches for the manipulation of lncRNAs with OMICS is indeed an important and promising approach. We have discussed it in section 2.2 (“lncRNA knockdown by CRISPRi”), line 408-415.
Round 2
Reviewer 3 Report
The revised version with added figures and edited text looks in good shape. I feel authors need to look carefully at the use of different abbreviations without expanding them. So, please add a 'Abbreviations section' to list different abbreviations used in the review manuscript.
Best,
Author Response
Thank you for your suggestion. We have carefully gone through the manuscript and defined all abbreviations that were not previously defined in the paper.